# Combining Software-Defined Radio Learning Modules and Neural Networks for Teaching Communication Systems Courses †

**Luis A. Camuñas-Mesa** *[ID] and **José M. de la Rosa** *[ID]

Instituto de Microelectrónica de Sevilla, IMSE-CNM, CSIC, Universidad de Sevilla, 41092 Sevilla, Spain
* Correspondence: camunas@imse-cnm.csic.es (L.A.C.-M.); jrosa@imse-cnm.csic.es (J.M.d.l.R.)
† This article is a revised and expanded version of a paper entitled Using Software-Defined Radio Learning Modules for Communication Systems, which was presented at the TAEE (Technology, Learning and Teaching of Electronics) Conference in Teruel, Spain, 29 June–1 July 2022.

**Abstract:** The paradigm known as Cognitive Radio (CR) proposes a continuous sensing of the electromagnetic spectrum in order to dynamically modify transmission parameters, making intelligent use of the environment by taking advantage of different techniques such as Neural Networks. This paradigm is becoming especially relevant due to the congestion in the spectrum produced by increasing numbers of IoT (Internet of Things) devices. Nowadays, many different Software-Defined Radio (SDR) platforms provide tools to implement CR systems in a teaching laboratory environment. Within the framework of a 'Communication Systems' course, this paper presents a methodology for learning the fundamentals of radio transmitters and receivers in combination with Convolutional Neural Networks (CNNs).

**Keywords:** cognitive radio; software-defined radio; communication systems; modulation; neural networks





## 1. Introduction

The integration of computing, communication, and information technologies into everyday objects, or things, is made possible by the miniaturization of microelectronics. Artificial intelligence (AI), big data, robotics, cloud computing, and IoT are among the disruptive technologies that are increasingly influencing our day-to-day lives. The demands brought on by the COVID-19 pandemic, during which millions of people were forced to interact virtually, have prompted the acceleration of this penetration. In fact, a set of digital layers surround our natural environment, enabling us to experiment with virtual entities and objects in an augmented reality known as the *metaverse* [1–3].

IoT is one of the most important technological players in the current digital transformation of our society. IoT refers to the interconnection of billions of *cyberphysical* entities, which can be real, virtual, or use a hybrid software/hardware structure. These *cyberphysical* entities are able to communicate with one another, sometimes without the need for human intervention, thanks to machine-to-machine communication protocols. In addition, it is anticipated that IoT technologies will have a potential impact on the global economy of USD 11.1 trillion by 2025, which equates to more than 10% of the world's gross domestic product. There are approximately 30 billion connected devices as of 2023, and this is expected to rise to 350 billion by 2030 [4].

The widespread utilization of electromagnetic spectrum is one of the main outcomes of the IoT boom. Even though new frequency bands such as millimeter wavelengths (mm-Wave) are being included in the most recent 5G and 6G mobile networks generations, data traffic continues to grow [5–10]. By dynamically modifying the transceiver specifications in response to the information that is sensed from the electromagnetic environment, so-called CR [11] enables communication systems to make better use of the frequency spectrum.

Nonetheless, the viable execution of CR presents an enormous number of constraints. These include aspects connected to legal regulations associated with bands being licensed to mobile and internet services along with countless technical difficulties. Examples of the latter include the interference brought about by CR clients as well as spectrum management tasks (access, sensing, band allocation, hand-off, etc.) which must be carried out simultaneously with the signal processing [12–14].

In order to dynamically select the optimal set of performance metrics and transmission bands, CR-based terminals may benefit from embedding AI engines in their main subsystems. However, this necessitates new circuits and systems strategies with a high degree of programmability and reconfigurability [1,6,10,14–16]. In this manner, a few recent works have proposed utilizing AI (Machine Learning (ML) and Deep Learning (DL)) methods to improve the management of the electromagnetic spectrum and to facilitate the signal processing and performance of IoT nodes equipped with CR technology [15,17–19].

As a result, the development of effective IoT devices necessitates multidisciplinary expertise in wireless communications, DL, and microelectronics. During the last decades, many different works have proposed multiple ways to use CR concepts and SDR platforms for educational purposes [20–25]. Inspired by these references, initial pedagogical efforts were made to introduce this topic in the framework of the Department of Electronics and Electromagnetism of the University of Seville through two different bachelor theses [26,27]. As a continuation of this preliminary work, a learning methodology based on the use of SDR platforms as a pedagogical tool to improve the quality of teaching and learning in telecommunications and electrical engineering is presented in this paper. In order to accomplish this aim and contribute to the field, two distinct case studies are presented here. These case studies use SDR/CR systems as an application scenario and provide students with the opportunity to acquire both fundamental knowledge and practical insight regarding wireless communications, DL, and signal processing. To illustrate how the presented strategy can be applied to a variety of educational contexts, experimental results are presented.

## 2. Materials and Methods

### 2.1. SDR-Based Learning Methodology

The mobile terminals prevalent at the beginning of the wireless telecom era were basic electronic devices that mostly only transmitted voice data. Nearly three decades later, 5G mobile telecom is gradually being implemented, reaching data rates of tens of gigabits per second and operating in multiple frequency bands from the sub-6 GHz band to the mm-Wave band [28,29].

#### 2.1.1. Towards Software Defined Radio

Today, the majority of wireless devices have very small radio modules; however, it is challenging to maintain and scale the mechanism when seeking to add new communication modes and services. Dedicated Radio Frequency (RF) chipsets are usually required each time a new communication protocol is created. On the other hand, the rate at which new features are incorporated into handheld devices is beginning to exceed the rate of package reduction and the trend toward Systems on Chip (SoC) that technology downscaling facilitates. Moving away from pure hardware-based devices and towards hybrid hardware/software-based ones is necessary to address this issue. An SDR system, as Mitola envisioned in 1995 [30], is a universal radio platform that can be programmed to steer any frequency band and process arbitrary communication protocols while guaranteeing privacy and security and providing the necessary service quality [31].

An ideal SDR transceiver, conceptually depicted in Figure 1a, processes all digital information using three main components: a Digital Signal Processor (DSP), Analog-to-Digital (A/D) interface, and antenna. As shown in Figure 1b, an effective interface between RF signals and digital data necessitates at least some analog signal conditioning circuitry,

meaning that this ideal implementation is not practically feasible due to the significant power consumption of A/D interfaces.

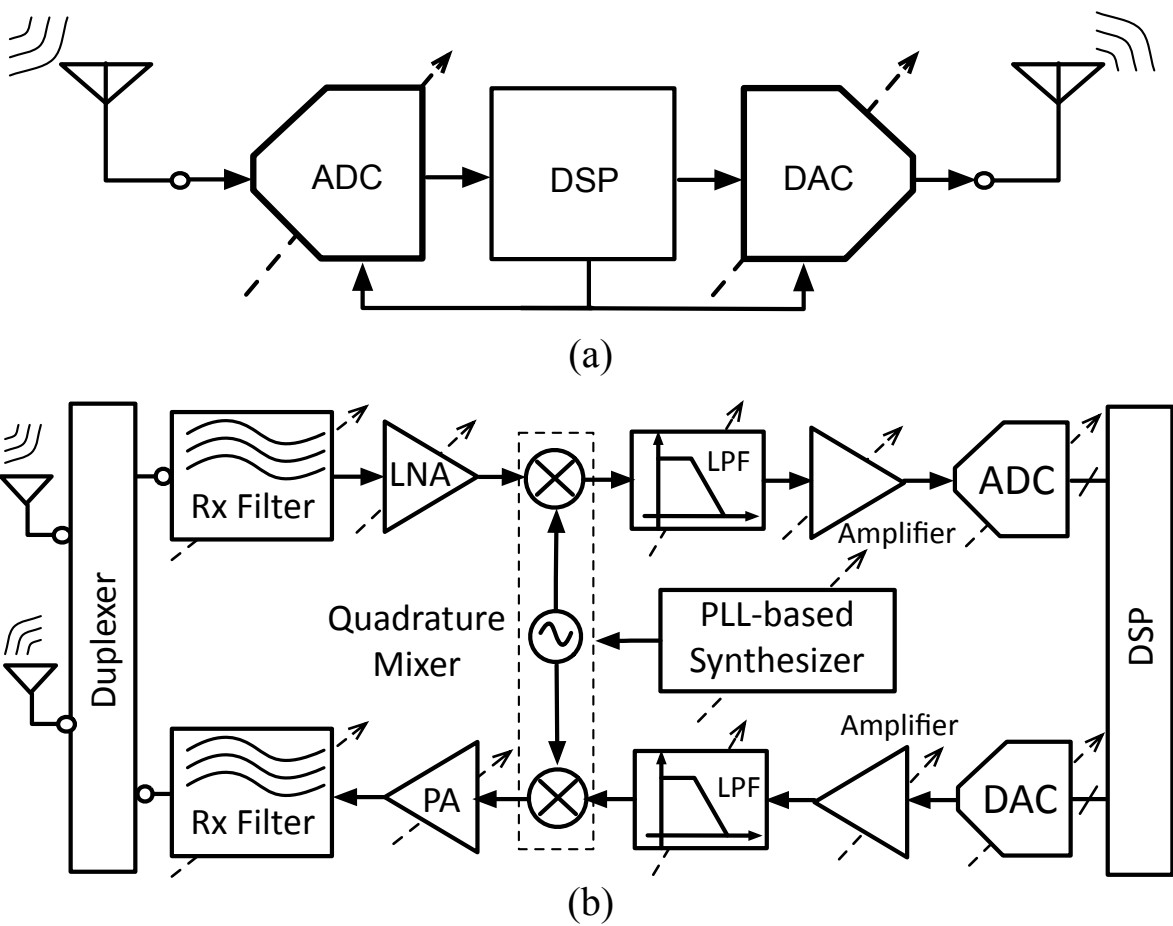

**Figure 1.** Block description of a generic SDR transceiver: (**a**) ideal blocks and (**b**) programmable multi-standard (direct-conversion) architecture.

2.1.2. Commercial SDR Boards as Learning Tools

The majority of signal processing is transferred to the digital domain by SDR transceivers. This feature makes software programming easier and allows SDRs to serve as the foundation for a variety of communication protocols and technologies, including CR; in essence, CR-based SDRs enable wireless networks and handheld terminals to dynamically utilize the RF spectrum. Consequently, both licensed and unlicensed spectrum can be utilized more effectively with lower power consumption and/or less interference. Along the same lines, SDR/CR-based mobile terminals can be prepared to *dynamically sense* the spectral environment and take advantage of received data to change their transmission/reception parameters on the fly, thereby improving the communication link and decreasing the amount of energy consumed.

SDRs' high level of programmability and adaptability can be used in an educational context to teach students the fundamentals of communication systems with real hardware and RF signal processing. To this end, Analog Devices ADALM-Pluto, Ettus USRP B210, Nuan BladeRF, and a number of other commercially available SDR platforms are offered by vendors. The primary specifications of these SDR boards can be found at [32], and they include, among other things: frequency tuning range, duplex method, Analog-to-Digital Converter/Digital-to-Analog Converter (ADC/DAC) resolution, FPGA (Field Programmable Gate Array) chipset, Rx (receiver) noise figure, etc. The majority of SDR boards' performance, at a reasonable hardware cost, can be very useful for teaching and learning.

The SDR-based learning module setup used in this work is depicted in Figure 2. A commercial SDR board such as the Analog Devices ADALM-Pluto, Nuan BladeRF, or Ettus USRP B210 is connected to a personal computer, where it can be controlled by various programming applications (for example, GNU Radio, URH, Gqrx, etc.) and programming languages (C, Python, and MATLAB).

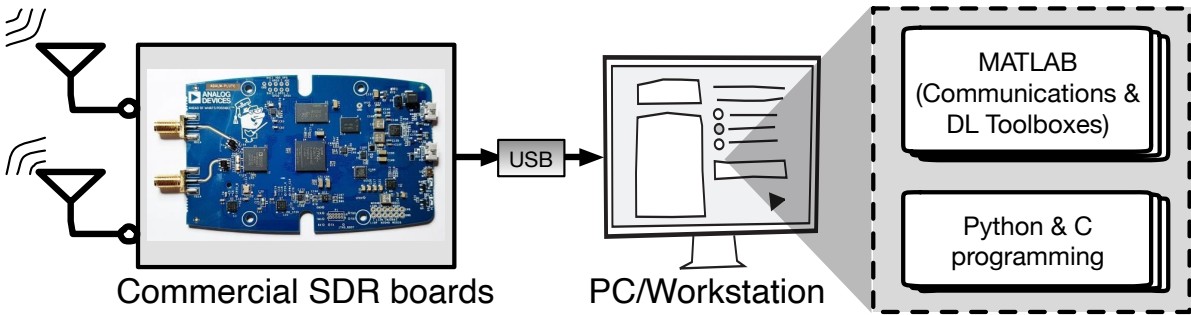

**Figure 2.** Proposed setup for a learning module based on commercial SDR boards.

### 2.2. Neural Networks for Cognitive Radio

Several of the tasks performed by CR systems require abilities that resemble those present in the human brain, suggesting that the use of neural networks can be especially suited to implementation of CR systems due to the similarities between artificial neural networks and the brain [17]. These required abilities are:

- The ability to deal with incomplete or erroneous data without affecting the results.
- The ability to process large amounts of data, given the massively parallel architecture.
- The ability to make decisions based on the processed data.

In this section, we briefly describe the main reasons why neural networks are so powerful, and summarize the most important applications of neural networks to CR systems.

#### 2.2.1. Neural Networks

Based on our understanding of the way in which the human brain processes complex information through a structure formed by layers of neurons [33], preliminary computing systems have been designed inspired by biological architecture, such as using the weights of the connections to implement learning mechanisms [34]. After the idea of backpropagation was introduced as a systematic method to train neural networks [35], it was demonstrated that these bio-inspired processing systems could be used for recognition applications [36], enabling the impressive growth of the field of neural network research observed during the last decades.

The generic structure of a neural network is shown in Figure 3a. Inspired by the human brain, the processing capabilities are distributed in very simple processing units (neurons) organized in different layers. In principle, each layer receives inputs from the previous layer and sends outputs to the next layer (this is the definition of a feed-forward neuron; recurrent networks include backward connections as well). The basic computational model of a neuron is illustrated in Figure 3b. In general, it is a processing unit with multiple inputs and single output. The operation performed by neuron $j$ is a weighted addition of inputs $x_i$, with the weights $w_{ij}$ applied to an activation function $\phi$ through a threshold $\theta_j$. Synaptic plasticity allows the weights of the network to be modified, supporting the implementation of learning capabilities through learning algorithms.

Under this generic definition of a neural network, the field has evolved towards a great number of specific implementations for many applications. In the next subsection, we focus on several applications of neural networks to CR.

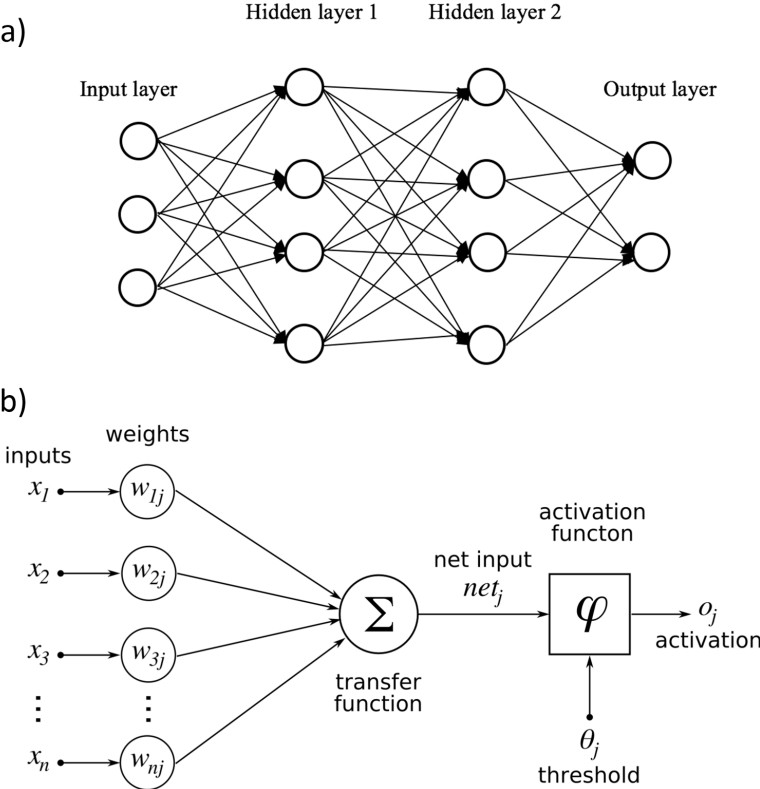

**Figure 3.** (**a**) Generic structure of a neural network and (**b**) basic computational model of a neuron.

### 2.2.2. Applications to Cognitive Radio

The use of neuromorphic techniques within the framework of CR systems can be approached from many different points of view. We first briefly review a number of the most relevant strategies for applying neural networks at the different layers of the protocol stack in a communications system [17]:

- In the physical layer, neural networks can be used for interference alignment, to classify the modulation modes, or to design efficient error correction codes.
- In the data link layer, neural networks can be used for resource allocation or link quality evaluation.
- In the network (or routing) layer, they can help to seek an optimal routing path.
- In higher levels, such as the application layer, they can be used to enhance data compression and multi-session scheduling.
- Outside of the protocol stack, there are many advantages for using neural networks in other functions, such as security and privacy protection.

One of the communication systems tasks in which neural networks can be used is associated with channel resource allocation, that is, dynamically selecting the most appropriate channel and modifying the parameters of the receiver. Many different approaches have been studied to solve this task without neural networks, for example, channel-hopping blind rendezvous protocols [37] and channel-hopping sequences in the SDR transmitter and receiver, including experiments implemented on Pluto boards [38]. In this section we focus on the application of neural networks for CR. In [19,39], Long Short-Term Memory (LSTM) networks were used to predict the future evolution of the occupation of different frequency bands. Using real-time measurements of the occupation for each band, one LSTM network per channel was used to predict their future evolution, then a decision block used the predicted signals to dynamically select the best band while tuning the receiver filter for that frequency.

Another application involves spatiotemporal modelling for traffic prediction. In this example [40], a dataset was collected from a large mobile network in a Chinese city that used nearly 3000 Base Stations. This dataset was then used to analyze the temporal and spatial correlations among neighboring Base Stations. A hybrid neural network model was proposed for spatiotemporal prediction, including an autoencoder-based model and LSTM networks for temporal modelling. The autoencoder model consisted of a Global Stacked Autoencoder and multiple Local Stacked Autoencoders, which was found to offer a good representation of input data, reduced model size, and support for parallel training.

One more example illustrates the use of a supervised Deep Neural Network (DNN) system for optimization of routing strategy in heterogeneous networks. This case [41] involved a heterogeneous network including both wireless and wired connections. The routing strategy was formulated as a combinatorial optimization problem, that is, a shortest-path routing problem in terms of graphs. In this work, the authors proposed a DNN that receives the input traffic pattern of the router and provides the desired output for network traffic control, i.e., the routing paths, showing very good performance.

Finally, one last application of neural networks to CR deals with modulation classification. In this example [18], a convolutional architecture consisting of two convolutional layers and two dense fully connected layers was proposed, for which several synthetic datasets were generated. The most difficult dataset included up to 24 different modulations, with different noise levels and over-the-air (OTA) transmission channels. These labeled signals were used to train different CNNs, and their performance was characterized for different values of Signal-to-Noise Ratio (SNR). This application is used further as a reference to describe the proposed Case Study 2 below.

### 2.3. Case Study 1: Communication System

In the framework of a 'Communication Systems' course, we have first proposed a practical methodology to acquire a basic knowledge about radio transmitters and receivers while performing practical experiments using commercial SDR modules specially suited to implement CR applications. This course is part of a Degree in Telecommunications Engineering, and is taught to fourth-year undergraduate students who have previously acquired a background in telecommunications and modulation schemes. Initially, we propose a case study which deals with the implementation of a communications system based on QPSK (Quadrature Phase-Shift Keying) modulation using two ADALM-Pluto SDR boards [42] controlled by MATLAB Simulink, as described in a Simulink example published on the MathWorks website [43]. The students are expected to reproduce this example using the available models, modify the parameters as needed, and characterize the quality of the communication. This experiment was designed to be performed in the lab with the available Pluto boards, with the students working in groups of two. The setup illustrated in Figure 4 is to be followed by each group, as follows:

- Student A implements the QPSK Transmitter module from a PC connected to the Pluto board.
- Student B implements the QPSK Receiver module from a PC connected to the other Pluto board.

The experiment's ultimate objective is to establish a communication link between the two students, send fixed configuration messages that the receiver can use to assess the quality of the communication, and adjust the transmission frequency to avoid busy frequency bands, thereby imitating a straightforward CR strategy. The subsequent subsections provide further details.

### 2.3.1. QPSK Modulation

In this case study, we asked the students to reproduce a Simulink example [43] to implement a QPSK modulator and demodulator in the transmitter and receiver blocks, respectively, despite the fact that this setup can be used to study and characterize various

modulation schemes. Utilizing a variety of characteristics of the carrier signal, it is possible to encode the transmitted bits using several kinds of digital modulation techniques:

- ASK methods (Amplitude-Shift Keying) rely on the amplitude.
- FSK methods (Frequency-Shift Keying) rely on the frequency.
- PSK methods (Phase-Shift Keying) use the phase.

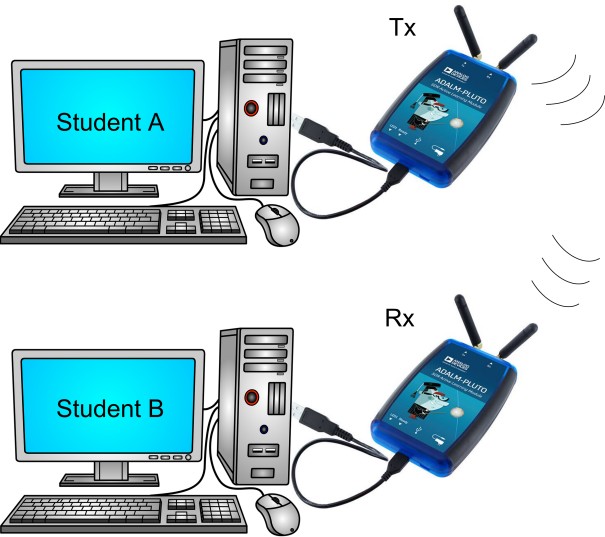

**Figure 4.** Description of the proposed experimental setup for Case Study 1.

In this work, we consider a particular kind of PSK modulation scheme, namely, QPSK (Quadrature Phase-Shift Keying), with four distinct phases equispaced, providing the chance to encode two bits for transmitted symbol. However, any other modulation can be used in the setup that is being proposed. In the years to come, we intend to introduce various modulations for use in the lab.

For the proposed modulation, the general expression for a QPSK signal is $s(t) = I(t)cos(w_c t) + Q(t)sin(w_c t)$, where $I(t)$ and $Q(t)$ represent the in-phase and quadrature components of the modulated signal. Figure 5 shows an illustration of an ideal constellation diagram with all four different transmitted symbols, with the I and Q signals represented on x and y axis, respectively. The figure shows that there is a 90° shift between adjacent symbols [44].

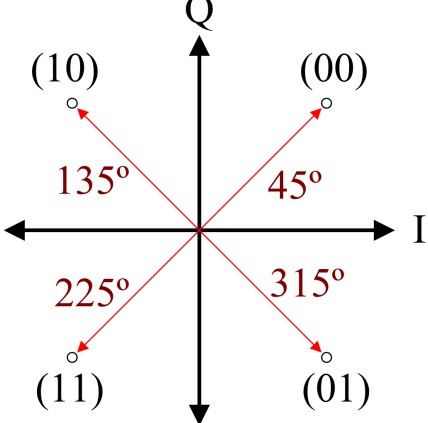

**Figure 5.** Typical constellation diagram for QPSK modulation.

2.3.2. Transmitter

MATLAB Simulink, which includes blocks designed to communicate with the ADALM-Pluto board, is used to physically implement both the transmitter and receiver modules.

The block diagram of the transmitter module available in this Simulink example [43] is shown in Figure 6. First, the "Bits Generation" block uses a 'Hello world' message as input and encodes it in bits. These bits are handled by the 'QPSK Modulator' in baseband utilizing Gray mapping. The block labeled 'ADALM-Pluto Transmitter' receives this baseband signal after it has been filtered and oversampled.

The main parameters which must be controlled by the students are:

- Radio ID: used to identify each Pluto board.
- Center frequency: the signal is modulated in baseband and afterwards translated to a certain transmission frequency, which must be within the range of 70 MHz to 6 GHz. This parameter should have the same value at both the transmitter and receiver in order to obtain the best possible quality of the communication.
- Gain: the attenuation of the signal while being transmitted, which must be with the range of −50 to 0 dB.
- Constellation ordering: the way in which the input bits are mapped into the QPSK constellation diagram.
- Phase offset: the phase associated with the first symbol in the constellation.

Constellation Diagrams for observing the transmitted symbols, Spectrum Analyzers for observing the frequency spectrum of the signals, and Time Scopes for observing the corresponding signals in time domain are among of the visualization blocks included in the module to enhance comprehension of the system, as depicted in Figure 6.

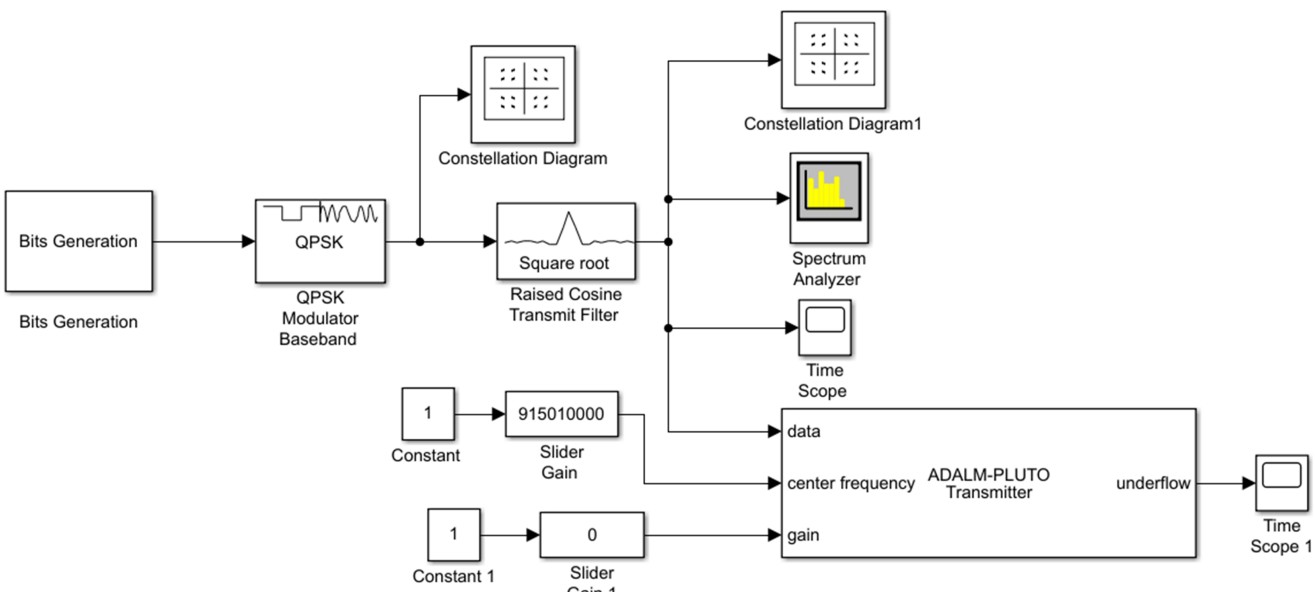

**Figure 6.** Transmitter block diagram implemented in Simulink.

### 2.3.3. Receiver

The 'ADALM-Pluto Receiver' block available in this Simulink example [43] is directly connected to the board Rx (receiver) antenna, while the 'QPSK Receiver' demodulates the obtained signal and calculates the Bit Error Rate (BER). This is shown in the block diagram of the receiver module in Figure 7. The red dashed rectangle below shows the complexity of the 'QPSK Receiver' block.

First, an AGC (Automatic Gain Control) is included in the proposed receiver block to maintain a constant phase gain and timing error and to stabilize the amplitude of the received signal. Next, the estimation quality is improved using an oversampling filter applied to the signal. Using correlation algorithms, the 'Coarse Frequency Compensation' system estimates and compensates for the frequency offset. The 'Image Synchronizer' then utilizes a PLL (Phase-Locked Loop) to reduce the mismatch related to the sampling rate

between transmitter and receiver. Using a different PLL, the 'Carrier Synchronizer' creates a frequency compensation that is more precise. The 'Preamble Locator' is then utilized to recognize the header included in the transmitted message, and the 'Data Decoding' block demodulates the message.

As in the transmitter framework, this module incorporates various types of visualization blocks (Constellation Diagrams, Time Scopes, and Spectrum Analyzers) to assist students in fine-tuning the demodulator, debugging it, and better better understanding its behavior, which is of course the primary goal of this learning experience.

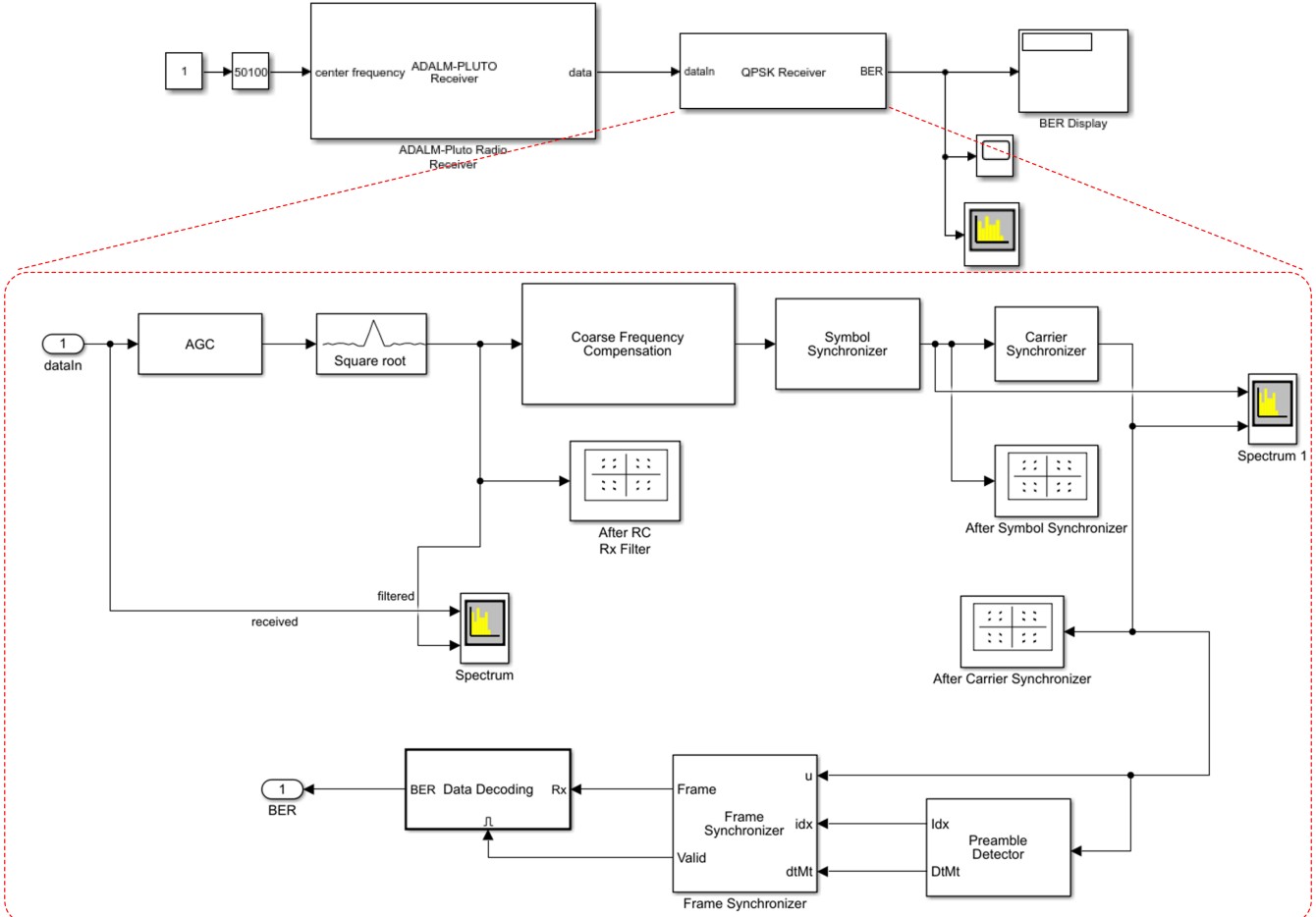

**Figure 7.** Receiver block diagram implemented in Simulink.

## 2.4. Case Study 2: Modulation Recognition

In the same framework described before, we define a second case study based on the combination of the proposed SDR boards with the implementation of neural networks capable of identifying the modulation of the received signal [18]. Although the targeted students may not necessarily have a deep background in neural networks, we consider this case study to be a good opportunity for them to become familiar with the main basic concepts from a practical point of view just by following certain guidelines. The goal of this experiment is to transmit signals using different modulation schemes and implement a neural network at the receiver that performs a classification task to identify the modulation. From a teaching perspective, this experiment can be very useful to illustrate an important aspect of CR systems, namely, that the receiver has to re-configure automatically in real time in order to be able to demodulate signals using different modulation schemes (although a deeper analysis of these modulation schemes is beyond the scope of this work). Thus, we

propose the lab setup indicated in Figure 8, where two PCs are used for different purpose to implement the whole communication system:

- One PC, denoted Tx, is responsible for loading a dataset consisting of a group of signals modulated using different schemes (available at [45]) and transmitting these signals using a Pluto board configured as transmitter.
- Several PCs (one for each student), denoted Rx, are responsible for receiving the transmitted signals using a Pluto board configured as a receiver; they implement a neural network in the form of software that can process the received signals and identify the modulation used for each signal. Initially, the neural network is characterized by processing the dataset locally (i.e., without using the Pluto board), and finally the whole system is characterized using Tx and Rx i (where i represents each student) through the corresponding Pluto boards.

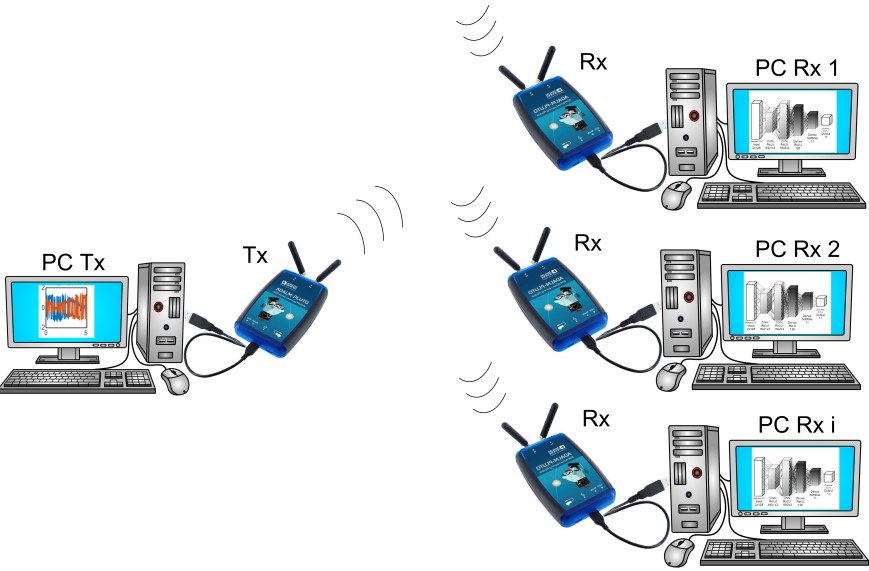

**Figure 8.** Illustration of the experimental lab setup used in Case Study 2.

Both Tx and Rx use MATLAB, which provides with Simulink blocks for interfacing with the Pluto board (as shown before) as well as with a visual tool called deepNetworkdesign, which allows Deep Neural Networks (DNNs) to be implemented on Rx using predefined available blocks. This case study can be described from two different perspectives, namely, the dataset transmitted through the SDR boards, and the Convolutional Neural Network used to identify the modulation scheme, which is the main goal of this case study.

### 2.4.1. Dataset

For the dataset transmitted in this experiment, we used the random data modulated for transmission made available by O'Shea et al. [45]. Initially, we considered eleven different modulation schemes while making a distinction between analog and digital modulations. Analog modulation involves continuous carrier signal with the ability to modify a number of its properties over time to represent the information signal. In digital modulation, a discrete carrier signal carries binary information related to the information signal. Here, we briefly describe the eight digital and three analog modulations considered in the dataset.

Digital modulations:

- QPSK, BPSK, and 8PSK.
  These three modulation schemes belong to the PSK category, in which the phase of the carrier can take several different values from a given discrete subset, making for a limited number of available states. Depending on the number of available phases, it is possible to obtain BPSK (two phases), QPSK (four phases), or 8PSK (eight phases).

- GFSK and CPFSK.
  The second category of digital modulation is FSK, in which two or more frequencies are used to encode each symbol. Here, we consider two different types: GFSK (Gaussian Frequency Shift Keying), where data pulses are first filtered by a Gaussian filter after which a logic 1 is represented by an increment of the carrier frequency and a logic 0 by a decrement of the carrier frequency; and CPFSK (Continuous Phase Frequency Shift Keying), where the phase is continuous, which is desirable for transmission over band-limited channels.

- 16QAM and 64QAM.
  These schemes belong to the category of Quadrature Amplitude Modulation (QAM), in which two carrier waves are used with the same frequency while being out of phase with each other by 90º in a condition known as quadrature. The transmitted signal is obtained by adding the two carrier waves together. The input flow of the digital bit streams can be divided into groups of bits needed to generate N different modulation states. In this case, we consider two possible values of N: 16QAM and 64QAM.

- PAM4.
  The last category of digital modulation is Pulse Amplitude Modulation (PAM), in which the phase and frequency are fixed and the amplitude changes. Different PAM schemes can be obtained depending on the number of possible values for the amplitude that the carrier wave can take. In this case, we use $N = 4$ (PAM4).

  Analog modulations:

- B-FM.
  The first category of analog modulation considered in this dataset is Frequency Modulation (FM), in which the information is encoded in the carrier wave by varying its instantaneous frequency.

- AM-SSB.
  In a second analog category, we consider Amplitude Modulation (AM), in which the information is encoded in the carrier wave by varying its instantaneous amplitude. In particular, we first focus on Single-Sideband Modulation (SSB), which reduces transmission power and bandwidth by sending only half of the bandwith originally generated by AM modulation.

- AM-DSB.
  Finally, an additional case of AM modulation is Double-Sideband, which does not implement the power and bandwidth reduction, as the whole modulated signal is transmitted. The main difference from basic AM is that AM-DSB does not include carrier re-insertion.

A visualization of the signals generated using these eleven modulation schemes is presented in Figure 9, where they are represented in the time domain, and in Figure 10, where they are represented in the frequency domain.

Although it would be very interesting to implement all these different modulation schemes in the setup proposed in Case Study 1, that would be beyond the scope of this work in light of the limited time available in our lab sessions.

### 2.4.2. Convolutional Neural Network

As mentioned before, the goal of this case study is to implement a neural network in MATLAB that can process the signals received from the Pluto board and identify its modulation scheme. As a starting point, we considered the CNN proposed by O'Shea [45] represented in Figure 11a. This is a four-layer network with two convolutional layers followed by two dense fully-connected layers. All of the layers use rectified linear (ReLU) activation functions, with the exception of a SoftMax activation function in the one-hot output layer, which is the last layer and implements the classification task by activating one of the eleven outputs corresponding to the eleven modulation schemes.

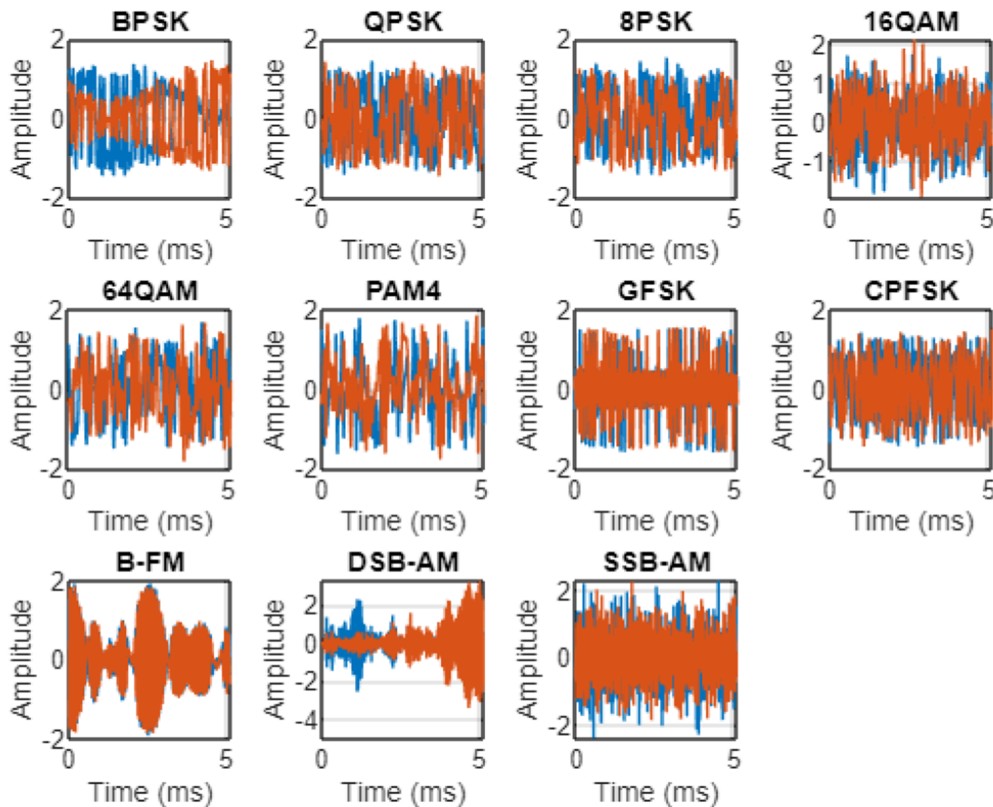

**Figure 9.** Representation of the eleven signals generated using all different modulation schemes in the time domain, as provided by O'Shea [45].

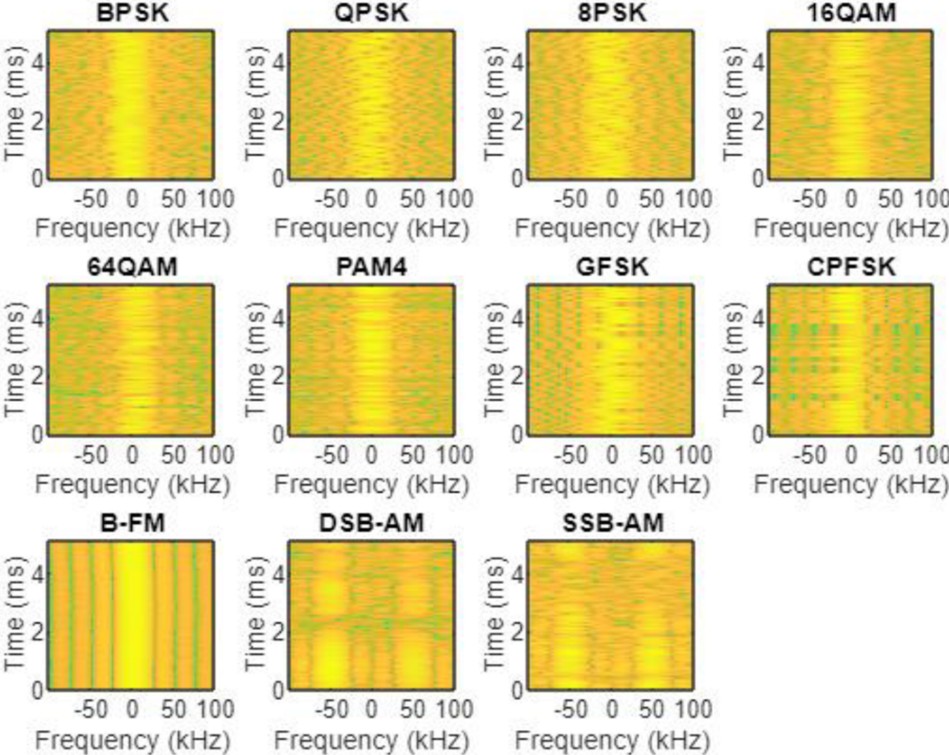

**Figure 10.** Spectrograms corresponding to the eleven different modulation schemes used in this case study, as provided by O'Shea [45].

Taking this CNN as a reference, we propose the implementation of similar networks in MATLAB using the deepNetworkdesign visual tool, with flexible parameters that can be modified by the students in order to maximize the performance in the experiment. Inspired by the network in Figure 11a, in order to include more flexibility in the lab experiment we implemented the CNN shown in Figure 11b, made up of six convolutional layers followed by a dense fully-connected layer and a SoftMax activation function in the final output classification layer. The main network parameters we propose for experimentation are as follows:

- Learning rate: this parameter represents the step of variation in the network weights in each iteration while training the network. A very small step makes the training process too slow, while a very large step leads to less optimal values for the weights.
- Number of layers: this parameter represents the number of convolutional layers in the network. In principle, deeper networks possess more powerful ability to learn more complex features, although an excessive number of layers can lead to an overfitting scenarios that reduce the performance of the network. Overfitting is an undesirable behavior that occurs when the network model provides accurate predictions for training data but not for new data, and can happen for several reasons, including insufficient training data, excessively noisy data (including irrelevant information), the model being trained for too long, or the model complexity being too high (learning the noise in the data).
- Number of filters: this parameter represents the number of neurons in the convolutional layers. To reduce the complexity of this case study, we propose using the same size in all convolutional layers and reducing the dimension directly in the last dense layers.
- Communication distance: this parameter does not modify the CNN, only the distance between both Pluto boards (i.e., the transmitter and receiver). The goal is to characterize the performance of the network when the communication distance increases.

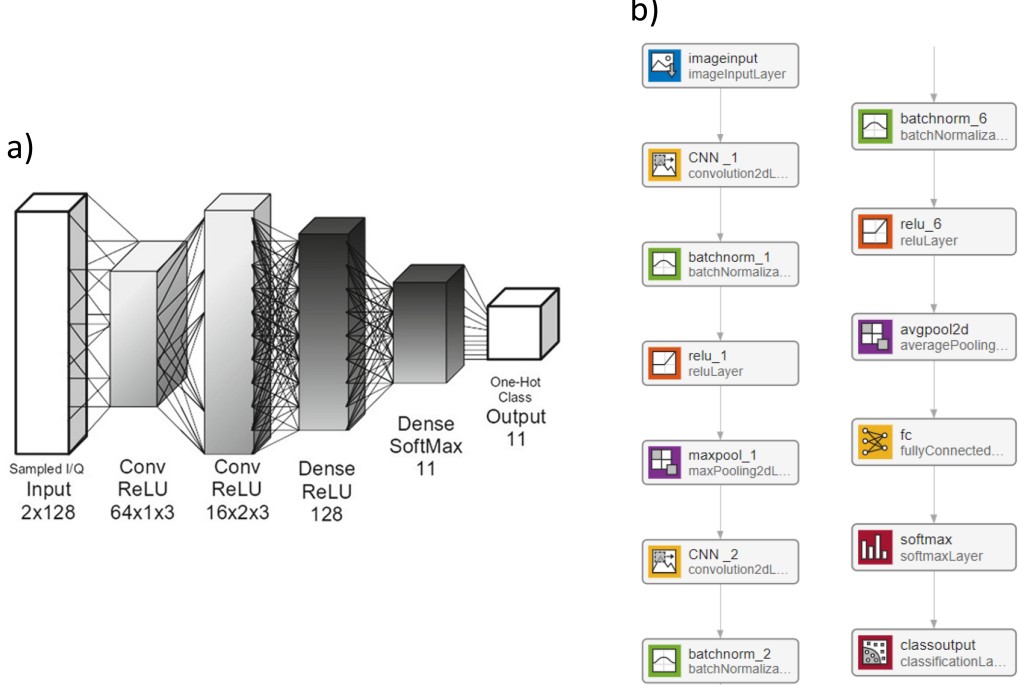

**Figure 11.** (**a**) Proposed neural network architecture for recognizing the eleven different modulations and (**b**) Simulink implementation of the network.

## 3. Results

### 3.1. Case Study 1

Several experiments were carried out with the setup depicted in Figure 4.

#### 3.1.1. Spectrum Visualization

In a first experiment, Universal Radio Hacker (URH) programming was utilized [46]. URH is a complete suite for wireless protocol analysis with native support for many common SDR systems, including ADALM-Pluto. By configuring this platform, the implemented system environment's radioelectric spectrum can be visualized and recorded. A message was continuously transmitted from the transmitter to the receiver using QPSK modulation at a center frequency of 1.66 GHz in the proposed setup (different bands were studied as well). The spectrum captured by the receiver is depicted in Figure 12 by a red trace, while the spectrum captured by the transmitter is depicted in black. While the desired signal has a distinct peak in its center frequency in the transmitter spectrum, the receiver spectrum contains a number of signals from other sources with distinct center frequencies that act as interference in this experiment. CR strategies that dynamically adjust the transmission frequency to compensate for this degradation are proposed, with the aim of examining how interference can lower communication quality.

#### 3.1.2. Frequency Deviation

In order to evaluate the significance of the various system parameters, a second experiment involves continuously transmitting 'Hello world' messages using QPSK modulation and configuring both the transmitter and receiver with MATLAB Simulink, as described in the preceding section. In particular, we focus on the center frequency of the transmitter ($f_{Tx}$) and receiver ($f_{Rx}$) modules in order to demonstrate the deterioration of communication brought on by even comparatively insignificant shifts in either frequency. Initially, the baseband spectrum of the transmitted signal in Figure 13a is obtained by configuring an ideal case with $f_{Tx} = f_{Rx} = 1.66$ GHz. After receiving and demodulating this signal, the constellation diagram in Figure 13b is produced. In all instances, the four distinct symbols $(00, 01, 10, 11)$ are easily discernible. Afterwards, $\Delta f = 1$ kHz is added to the receiver frequency $f_{Rx}$. The receiver constellation diagram in Figure 13c is the result of this minor misalignment between the transmitter and the receiver. Although the four symbols continue to be recognizable, certain measured points are assigned to the incorrect symbol, resulting in bit errors and incorrect message reconstruction.

The results shown in Table 1 were obtained by repeating the experiment with various frequency deviations and measuring the Bit Error Rate for each. While a perfect alignment has $BER = 0\%$, a small $\Delta f = 1$ kHz has $BER = 6.58\%$. Errors greater than 30% are caused by deviations greater than 2 kHz, indicating that even very small deviations make it impossible to demodulate the transmitted message correctly.

**Table 1.** Measured BER vs. frequency deviation.

| $\Delta f$ (kHz) | $BER$ (%) |
|---|---|
| 0 | 0 |
| 1 | 6.58 |
| 2 | 31.69 |
| 3 | 33.73 |

### 3.2. Case Study 2

Using the setup described in Section 2.4, several experiments were performed to analyze the effect of different parameters on the performance of the implemented neural network for modulation classification. Initially, the CNN was trained and tested directly with the dataset described before without using the Pluto board in order to optimize the parameters, then the whole system was tested with the data sent through the SDR boards.

In each experiment, all modulated signals were processed to obtain classification results and evaluated using the accuracy of the classification.

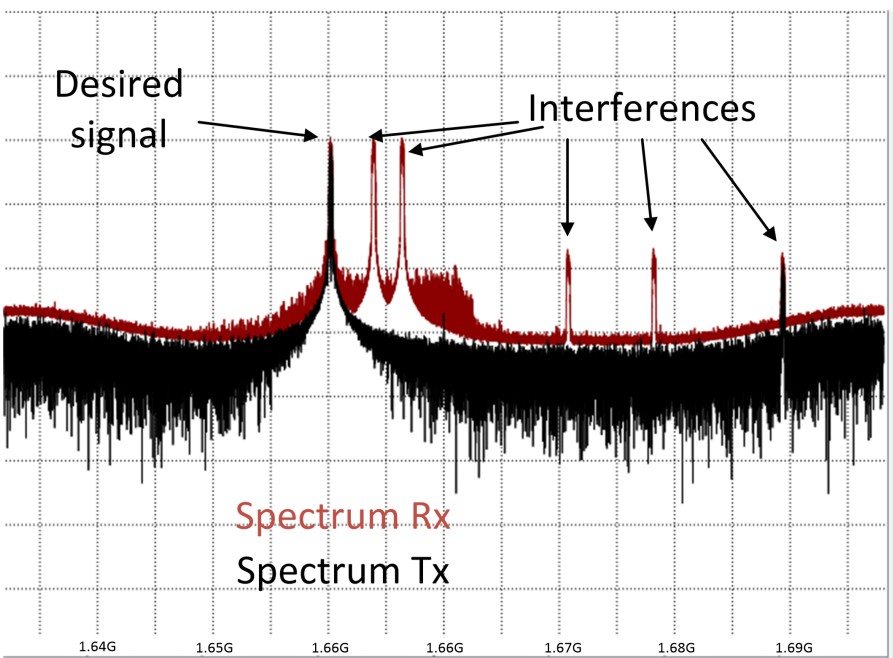

**Figure 12.** Representation of the measured frequency spectrum on the transmitter and receiver sides.

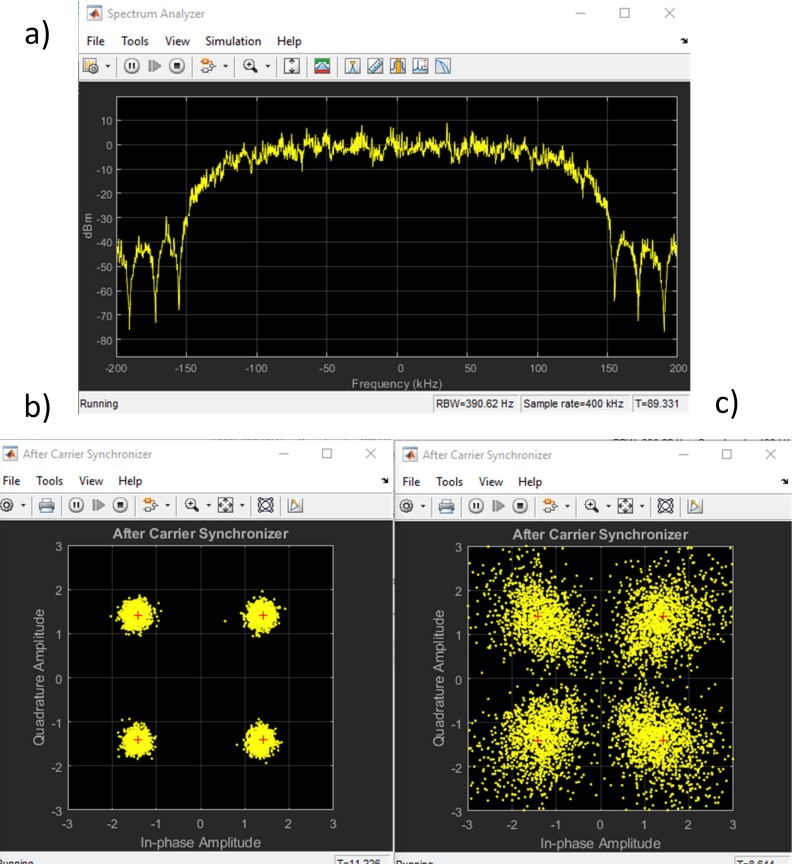

**Figure 13.** Measurements obtained from Simulink: (**a**) frequency spectrum of the transmitted signal in baseband, (**b**) constellation diagram obtained by the receiver when both boards are correctly aligned in frequency, and (**c**) constellation diagram when a small frequency deviation is introduced.

### 3.2.1. Effect of Learning Rate

For a first experiment, the CNN described in Section 2.4.2 was implemented in MAT-LAB with six layers and sixteen filters. The first parameter to be characterized was the learning rate, which defines the magnitude of change applied to the network weights during the training stage. If the learning rate is too small, the training algorithm evolves very slowly and requires a long time to converge to an efficient solution. If the learning rate is too large, the algorithm evolves faster; however, it might converge to an incorrect solution. In order to find an optimum value, the students were asked to train the network with different values of the learning rate and evaluate the performance of the CNN. Two measurements were used to characterize the performance:

- Training time: how long it takes the algorithm to converge.
- Accuracy: the percentage of input samples which are correctly classified by the network.

Table 2 includes the results obtained when applying different values for the learning rate; it can be seen that 0.01 provides the best accuracy (although very similar to 0.02) without leading to an important difference in the training time. Therefore, this is the value used in the other experiments. Although it might be expected that a larger training time would be obtained for lower learning rates, there are no important differences in the applied values. This might be due to the implementation of the training algorithm provided by MATLAB, which probably limits the training time even when it does not converge properly. For this reason, the accuracy degrades with a learning rate of 0.001 while having a very similar training time. However, the limited available time in the lab made it impossible to carry out any deeper analysis of the obtained results, as the main objective of this work was to help students develop their experimental skills while improving their knowledge of basic concepts, not to obtain the most precise results. Figure 14 illustrates the training progress for a learning rate of 0.01, where the horizontal axis represents the number of training iterations and the blue line indicates the accuracy. As can be seen, the network converges in less than 1200 iterations.

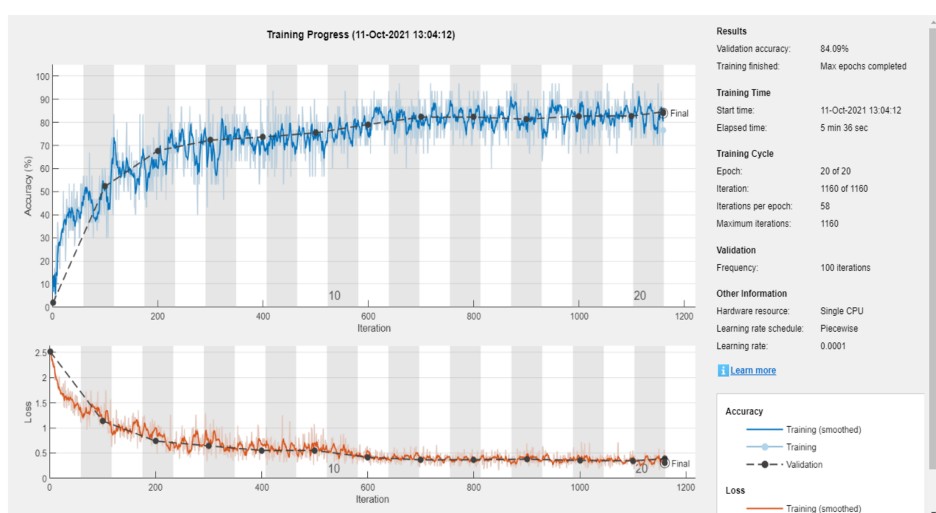

**Figure 14.** Training progress of the neural network with a learning rate of 0.01.

**Table 2.** Characterization of the effect of the learning rate for six layers and sixteen filters.

| Learning Rate | Accuracy | Training Time |
|---|---|---|
| 0.001 | 57.27% | 4 min 55 s |
| 0.01 | 84.09% | 5 min 36 s |
| 0.02 | 83.18% | 5 min 31 s |
| 0.1 | 70.45% | 4 min 55 s |

A very common way to represent the accuracy of a network on classification tasks in greater detail is the confusion matrix; each row in the matrix indicates the true class for each input, while each column indicates the predicted class, as shown in Figure 15. Ideally, this matrix should only present non-zero values on the principal diagonal, meaning that the predicted class is always the same as the true one. However, in practice we always obtain an error, resulting in a total accuracy lower than 100%. Figure 15a depicts the confusion matrix for a learning rate of 0.1, showing poor classification results for the 16QAM, 64QAM, 8PSK, B-FM, and QPSK modulations. Figure 15b shows the confusion matrix for a learning rate of 0.01, showing poor classification results for only the 16QAM and QPSK modulations.

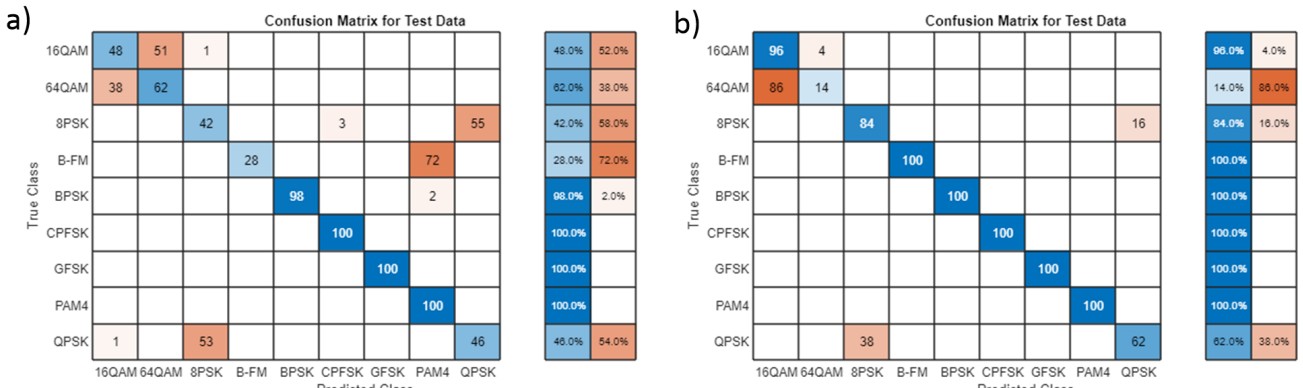

**Figure 15.** Confusion matrices obtained when classifying nine types of modulation with learning rates of 0.1 (**a**) and 0.01 (**b**).

### 3.2.2. Effect of Number of Layers

In a second experiment, we changed the number of layers in the network in order to obtain the best accuracy. Table 3 presents the characterization of this effect, where four layers results in the best performance in terms of accuracy (the training time is not as critical here, as the network has to be trained only once at the beginning, and the obtained values are all considered reasonable). These results show that increasing the number of layers is not always the best way to improve performance, as too many layers can produce an overfitting effect. Figure 16 shows the confusion matrix obtained when the number of layers is four, with only two of modulations presenting a recognition rate lower than 70%.

**Table 3.** Characterization of the effect of the number of layers for sixteen filters and a learning rate of 0.01.

| Number of Layers | Accuracy | Training Time |
| --- | --- | --- |
| 6 | 84.09% | 5 min 36 s |
| 5 | 90.00% | 10 min 30 s |
| 4 | 92.73% | 10 min 8 s |
| 3 | 89.41% | 9 min 54 s |

### 3.2.3. Effect of Number of Filters

In a similar way, we proposed optimizing the number of filters in the convolutional layers while keeping a total of four layers, obtaining the results presented in Table 4. In this case, it can be seen that a larger number of filters results in higher accuracy. However, when the number of filters is increased up to 64, we observe an important increment in the training time (>20 min); thus, it was considered impractical to continue increasing the number of filters.

Figure 17 presents the confusion matrix obtained for 64 filters, which provides the best performance. As can be seen, we obtain a recognition rate larger than 80% for all modulation types, and six of them reach 100%.

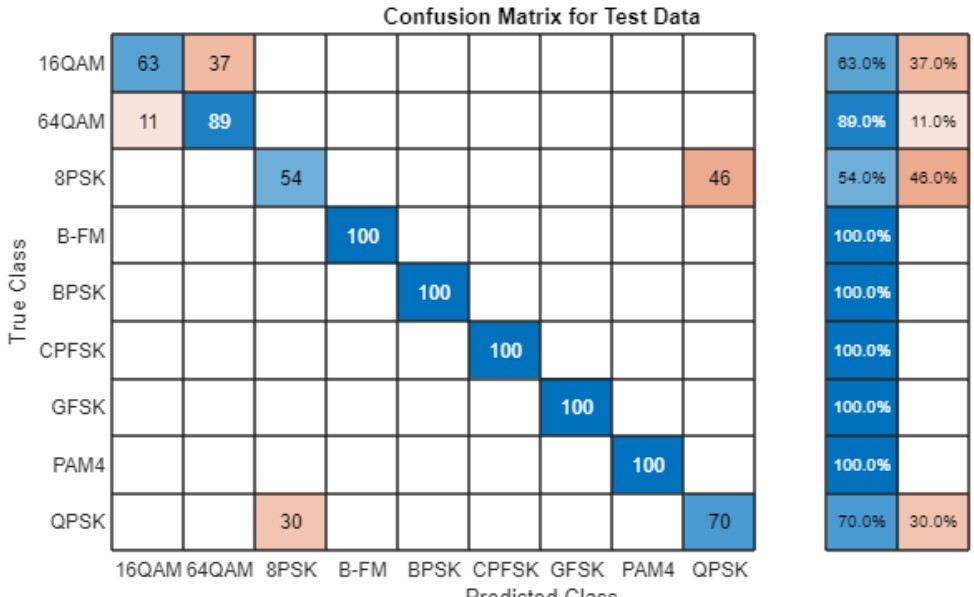

**Figure 16.** Confusion matrix obtained when classifying nine types of modulation with a neural network consisting of four layers.

**Table 4.** Characterization of the effect of the number of filters for four layers and a learning rate of 0.01.

| Number of Filters | Accuracy | Training Time |
| --- | --- | --- |
| 16 | 92.73% | 10 min 8 s |
| 32 | 93.64% | 10 min 20 s |
| 64 | 96.82% | 21 min 11 s |

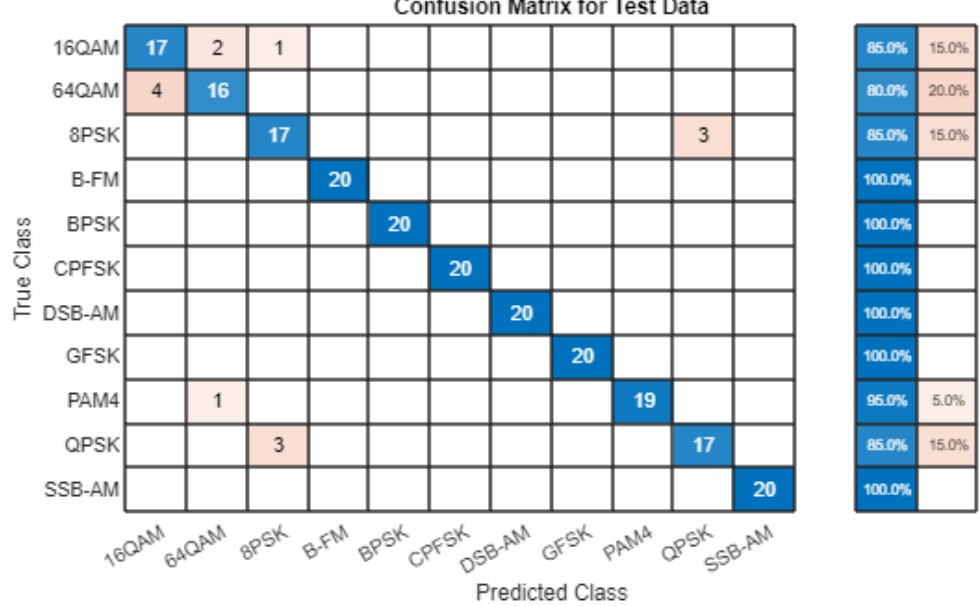

**Figure 17.** Confusion matrix obtained when classifying eleven types of modulation with a neural network consisting of four layers with 64 filters.

3.2.4. Effect of Communication Distance

As a final experiment, we proposed implementing the CNN together with the SDR platform in order to validate the complete setup proposed in Figure 8. For this experiment,

we used the original network model with six layers and sixteen filters; even though this network presents slightly lower accuracy, it can be trained faster, which is convenient for a repetitive experiment. We implemented the model on the receiver side, while the input dataset was sent from the transmitter side. The goal of this experiment was to measure the effect of the communication distance, i.e., the distance between the transmitting and receiving Pluto boards. For this, we established five different separations: 0 cm, 10 cm, 20 cm, 1 m, and 3 m. Larger distances could not be tested in the lab environment. For each separation, we repeated the experiment ten times in order to obtain more statistics. The results are depicted in Figure 18, where the error bars represent the dispersion of the measured accuracy. As can be seen, there is no clear influence of the distance on the mean accuracy, which is always around 85%, though larger separations produce a higher dispersion in the behavior of the system.

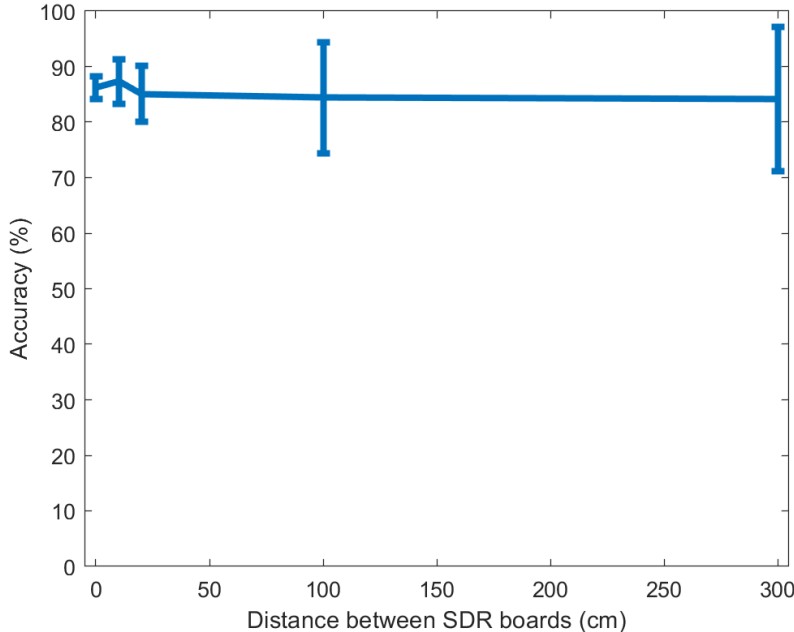

**Figure 18.** Effect of the distance between SDR boards on classification of the modulation type.

## 4. Discussion

In this paper, we propose two different experimental setups based on commercial SDR boards for use in practical lessons in the framework of a 'Communication Systems' course. These setups were especially appropriate for implementing basic approaches to CR systems, establishing radio connections between two users with dynamic parameters, and experimenting with important concepts associated with the software implementation of neural networks.

As a first case study, we implemented a transmitter–receiver system using QPSK modulation and characterized the quality of the communication. The results include observing the critical effect of interference when working in saturated frequency bands and the degradation in quality produced by misalignment between the transmitter and receiver frequencies.

In a second case study, we implemented a transmitter–receiver system with a CNN on the receiver side used to identify the modulation of the transmitted signal. This application is specially suited for CR, as it allows for dynamic reconfiguration in order to adapt to the specific modulation used by each signal.

The lessons learned from the presented education experience are that students become more motivated and satisfied than when following a traditional lab course, translating into higher grades on the part of a majority of students. From the limited experience obtained thus far, we have compared the average grades of students over the last five

years previous to implementing the proposed experiments to the one year completed incorporating the program presented in this paper. In previous years, the average grade in the 'Communication Sytems' course was 6.84, while the average grade when implementing the proposed methodology was 7.91, both on a scale from 0 to 10. We intend to continue monitoring these results in the coming years. At the end of each year, we collect anonymous surveys in which students can provide feedback about their perception of the course along with a final evaluation between 0 and 5. In the five previous years, the average evaluation was 4.1, while in the year after implementing the practical sessions we obtained an average evaluation of 4.7. We have found that as students keep working on the same project for several lab sessions, they develop a deeper interest in the topic and improve their ability to connect different theoretical concepts, especially those related to the implementation of both neural networks and communication systems.

**Author Contributions:** Conceptualization, L.A.C.-M. and J.M.d.l.R.; Methodology, L.A.C.-M. and J.M.d.l.R.; Software, L.A.C.-M.; Validation, J.M.d.l.R.; Investigation, L.A.C.-M.; Writing-original draft, L.A.C.-M. and J.M.d.l.R.; Writing-review and editing, L.A.C.-M.; Supervision, J.M.d.l.R.; Project administration, L.A.C.-M. and J.M.d.l.R.; Funding acquisition, J.M.d.l.R. All authors have read and agreed to the published version of the manuscript.

**Funding:** This work was supported in part by MCIN/AEI/10.13039/501100011033 under Grants PID2019-103876RB-I00, PID2022-138078OB-I00 and PID2019-105556GB-C31; in part by the European Union "ESF Investing in Your Future" and in part by "Junta de Andalucía", Spain, under Grant P20-00599 and PROYEXCEL-00060.

**Data Availability Statement:** No new data were created or analyzed in this study. Data sharing is not applicable to this article.

**Conflicts of Interest:** The authors declare no conflict of interest.

## Abbreviations

| | |
|---|---|
| 5G | Fifth Generation |
| 6G | Sixth Generation |
| 8PSK | Eight Phase-Shift Keying |
| A/D | Analog-to-Digital |
| ADC | Analog-to-Digital Converter |
| ADALM | Advanced Active Learning Module |
| AGC | Automatic Gain Control |
| AI | Artificial Intelligence |
| AM | Amplitude Modulation |
| AM-SSB | Amplitude Modulation with Single-Sideband |
| AM-DSB | Amplitude Modulation with Double-Sideband |
| ASK | Amplitude-Shift Keying |
| BER | Bit Error Rate |
| BPSK | Binary Phase-Shift Keying |
| CNN | Convolutional Neural Network |
| COVID-19 | Coronavirus Disease 2019 |
| CPFSK | Continuous-Phase Frequency-Shift Keying |
| CR | Cognitive Radio |
| DAC | Digital-to-Analog Converter |
| dB | Decibel |
| DL | Deep Learning |
| DNN | Deep Neural Network |
| DSP | Digital Signal Processor |
| FM | Frequency Modulation |
| FPGA | Field-Programmable Gate Array |
| FSK | Frequency-Shift Keying |
| GFSK | Gaussian Frequency-Shift Keying |
| GHz | GigaHertz |

| | |
|---|---|
| ID | Identification |
| IoT | Internet of Things |
| LSTM | Long Short-Term Memory |
| MHz | MegaHertz |
| ML | Machine Learning |
| mm-Wave | Millimeter-Wavelength |
| NN | Neural Network |
| OTA | Over-The-Air |
| PAM | Pulse Amplitude Modulation |
| PLL | Phase-Locked Loop |
| PSK | Phase-Shift Keying |
| QAM | Quadrature Amplitude Modulation |
| QPSK | Quadrature Phase-Shift Keying |
| ReLU | Rectified Linear Unit |
| RF | Radio Frequency |
| Rx | Receiver |
| SDR | Software-Defined Radio |
| SNR | Signal-to-Noise Ratio |
| SoC | System-on-Chip |
| Tx | Transmitter |
| URH | Universal Radio Hacker |
| USRP | Universal Software Radio Peripheral |

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
