# Peer review of "Combining Software-Defined Radio Learning Modules and Neural Networks for Teaching Communication Systems Courses†"

_information, doi:10.3390/info14110599_

Round 1

Reviewer 1 Report

Comments and Suggestions for Authors

In this paper, the authors presented a learning methodology of identifying the modulation schemes of transmitted signals using a QPSK-based SDR transmitter/receiver in combination with convolutional neural networks.  Two cases studies were presented and their results were provided. 

In Section2.2, the authors mentioned that “One of the tasks where Neural Networks can be used for communication systems is associated with channel resource allocation...”  There is another approach of channel resource allocation without the need of neural networks. This approach uses channel-hopping blind rendezvous protocols and channel-hopping sequences in the SDR transmitter and receiver. Experiments of implementing channel-hopping sequences in PlutoSDR and USRP had been published. This kind of work should be cited and mentioned in the paper.

The proposed methodology requires students with background in the theories of telecommunications and neural networks. It is unclear what kind of students will be targeted for and will benefit from the proposed methodology. What are their education background (computer science, electrical engineering, computer engineering, or others) and level (undergraduate or graduate)?

The first case study in Section 2.3 was copied from a PlutoSDR Simulink example in MathWorks’ website. However, the authors wrote this section as if the Simulink transmitter and receiver models were created by them. There was no citation of the source or proper credit provided.

The QPSK receiver involves complex and multiple levels of synchronization. Were students taught about the concept and functionality of each block before they tested the QPSK transmitter and receiver?  If not, how the students understand the results presented in Section 3.2?  Along this line, were the students explained the modulation schemes in Section 2.4.1, and their characteristics and pros/cons? 

Figure 4 and 8 are similar and contain no useful information. Instead of Fig. 8, the authors should provide complete Simulink diagrams of how different modulation schemes were implemented in PC1 and CNN and the deepNetworkdesign visual tool were implemented in PC2, as mentioned in p. 9 of the paper.  

In Section 2.4.1, different modulation schemes were mentioned. They should be tested in Section 3.1 to show the effects of different modulation schemes to BER and constellation diagram.

While different modulation schemes were included in the test results in Sections 3.2.1--3, there was not mentioned in Section 3.2.4. Did the authors obtain the same result for all these modulation schemes in Fig. 18?

Reviewer 2 Report

Comments and Suggestions for Authors

This paper is written as a lab tutorial, presenting some background information in the introduction (for a Special Magazine Issue "Technology, Learning and Teaching of Electronics with Information Applications". Thus, it's difficult to expect scientific findings. Some remarks regarding the content: 1) Discussion about some kinds of modulation and QPSK implementation in MATLAB with BER analysis depending on carrier frequency mismatch is rather basic but ok for a lab. 2) ML for automatic modulation classification part is a more interesting, hot topic. However, there are some noticeable flaws in explanations and obtained results. E.g., it is said twice that the smaller the learning rate, the slower the algorithm's convergence.  However, in Tab.2, these tradeoffs are unclear; the training time for rate 0.001 is less than for other rates. Why? Then, the effect of model overfitting is mentioned but not explained at all.   Writing and formatting: really poor. Below are just some examples: 1) p.1: IoT abbreviation introduced 3 times (IoT(Internet of Things) on line 4, Internet of Things (IoT) on lines 15 and 21). Then, after being introduced 3 times, it's used in full form again on line 22. 2) Same with Cognitive Radio (CR). See lines 1 and 35. Then, it's used again in full form on p.3 and several places later. 3) Millimeter waves: introduced as mm-Wave on p.1, then used as mm-W on p.2. 4) Artificial Intelligence (AI): introduced on p.1, then again on p.2. Same with SDR. 5) p.5: autoencoder, Auto-Encoder, Aut-Encoders... 6) Not introduced: ML, A/D (p.2), FPGA, ADC/DAC (p.3)... 7) p.2: MATLAB, p.11: Matlab. And so on...

Comments on the Quality of English Language

There are no specific comments on the quality of English language.

Reviewer 3 Report

Comments and Suggestions for Authors

The authors present two modules for the teaching of cognitive radio principles within a Communication Systems course. Using SDR and related technologies (such as CR) is a necessary and growing trend (certainly not “novel methodology”, line 187). As such, papers that report on this pedagogical need and provide modules that can be adopted by others are of importance.

The largest issue with this paper is that, given this framework, it includes no references or background material on other educational efforts in pedagogy, particularly how that drove the educational design of the modules. In addition, the concluding paragraph “The lessons learnt…” is provided with no supporting evidence, methodology, validity of results, etc. For a paper that purports to be about education, these omissions are glaring. The paper requires a major revision before acceptance to include this context and fully supported educational outcomes.

Other educators would appreciate access to the modules, so please consider providing this.

The paper is fairly well written otherwise, with small English usage issues (e.g., line 5 “Defined”; line 6 remove “with”; line 14 “downscaling” à “miniaturization”; line 47 “late” à “recent”; line 91 “dynamically”; line 199 “link of communication” à “communication link”; etc.)

Some particular elements that could use clarification:

Line 16-18: Not clear why COVID-19 drives those disruptive technologies, seems the be a statement to hit on current buzzwords.

Line 27, is this USD?

Line 61: while voice was dominant, there were other data streams that mobile terminals could transmit

Line 76: it is not clear what is meant by “to steer”

Line 109: there are many arguments that CR (as currently implement) is not using “certain abilities present in the human brain”à my cognitive science friends would certainly argue CR does not.  So, substantiate this claim.

Line 193: it is not clear why the “experiment must be performed in the lab”, could the ADALMs not be hooked up to laptops and transported elsewhere?

Around line 296: How are nonidealities (noise, phase delay, etc.) added to the signals.  Also, how does the system work with Modulations that are not encoded into the system?  Does it try to fit only to what it knows?

Figure 10: very difficult to see spectra due to colors chosen

Comments on the Quality of English Language

see above

Round 2

Reviewer 1 Report

Comments and Suggestions for Authors

no further comment.

Reviewer 3 Report

Comments and Suggestions for Authors

comments adequately addressed